# Increased Serum IgG4 Associates with Asthma and Tissue Eosinophilia in Chronic Rhinosinusitis Patients

**DOI:** 10.3390/pathogens9100828

**Published:** 2020-10-10

**Authors:** Mahnaz Ramezanpour, Hua Hu, Aden Lau, Sha Liu, April De Silva, Harrison Bolt, Karen Patterson, Maureen Rischmueller, Alkis J Psaltis, Peter-John Wormald, Susan Lester, Sarah Vreugde

**Affiliations:** 1Department of Surgery-Otolaryngology, Head and Neck Surgery, The Queen Elizabeth Hospital, Woodville SA 5011, Australia; mahnaz.ramezanpour@adelaide.edu.au (M.R.); 052101059@fudan.edu.cn (H.H.); adenhyl@hotmail.com (A.L.); sha.liu@adelaide.edu.au (S.L.); april.desilva@student.adelaide.edu.au (A.D.S.); alkis.psaltis@adelaide.edu.au (A.J.P.); peterj.wormald@adelaide.edu.au (P.-J.W.); 2School of Medicine, Faculty of Health Sciences, University of Adelaide, Adelaide SA 5005, Australia; Maureen.Rischmueller@sa.gov.au (M.R.); Susan.Lester@sa.gov.au (S.L.); 3Department of Surgery-Otolaryngology, Head and Neck Surgery, Shanghai General Hospital, Shanghai Jiaotong University, Shanghai 200240, China; 4Medicine Department, College of Medicine and Public Health, Flinders University, Adelaide SA 5042, Australia; harrison.bolt@uqconnect.edu.au (H.B.); karen.patterson@flinders.edu.au (K.P.); 5Rheumatology Department, Discipline of Medicine, The Queen Elizabeth Hospital, Woodville SA 5011, Australia

**Keywords:** chronic rhinosinusitis, immunoglobulin, IgG4, asthma

## Abstract

Chronic Rhinosinusitis (CRS) is a multifactorial disease where microorganisms’ innate and adaptive immunity can play a role. This study assessed the total IgG, IgG subclasses, IgE and IgA levels in serum samples from CRS and non-CRS control patients in relation to the disease severity, phenotype, histopathology and comorbidities. Total serum IgG, IgG1, IgG2, IgG3, IgG4 and IgE was determined from 10 non-CRS controls, 10 CRS without nasal polyp (CRSsNP) and 26 CRS with nasal polyp (CRSwNP) patients using ImmunoCap assays. Tissue lysates were analyzed for IgG levels by ELISA. Immunohistochemical analysis was used to measure the expression of IgE and IgG4 in tissue sections. The presence of anti-nuclear antigens (ANAs) against 12 autoantigens in sera and tissue lysates was determined by immunoblot assays. Total serum IgG/IgG1/IgG2 levels were higher in CRS patients vs. controls (*p* < 0.001), but were not different between CRSwNP and CRSsNP patients (*p* = 0.57). Serum IgG4/IgE levels were increased in CRSwNP patients compared to controls (*p* = 0.006), however, this relationship was attenuated by the inclusion of covariates. Serum IgG4 levels were more strongly associated with asthma (*p* = 0.038, exact median test) and tissue eosinophilia (Spearman’s rank rho = 0.51, *p* = 0.016) than IgE levels. No systemic ANAs were detected in any of the subjects tested. There was a polyclonal increase in serum immunoglobulins in CRS patients with elevated IgG4/IgE levels in CRSwNP patients having tissue eosinophilia and asthma.

## Letter to the Editor

Chronic Rhinosinusitis (CRS) is characterized by persistent inflammation of the paranasal sinus mucosa associated with chronic relapsing infections. CRS is phenotypically classified into CRS with nasal polyps (CRSwNP) and CRS without nasal polyps (CRSsNP) based on the endoscopic evidence of nasal polyps in the middle meatus [1]. These two CRS phenotypes often show different immunopathological characteristics with Type 2 polarization more frequently found in CRSwNP patients. Previous studies have shown increased frequencies of B-cell subtypes and plasma cells along with increased IgA, IgG and IgE concentrations in CRSwNP tissue compared to controls [2,3]. Similar increases in immunoglobulin levels were not observed, however, in the serum of those patients [4]. Moreover, increased levels of nuclear-targeted autoantibodies within CRSwNP tissue have been shown particularly in severe recalcitrant CRS patients [5]. Whilst serum autoantibody levels were not tested, those findings indicate the potential of an autoimmune component that might contribute to the persistent inflammation in CRSwNP patients. This study investigated the tissue and serum autoantibodies and total IgG, IgG subclasses and IgE levels in CRS and non-CRS control patients in relation to CRS disease severity, phenotype, histopathology and comorbidities. Materials and methods can be found in this article’s online repository. Forty-six patients (age range 19–80 years, 35% female) were included in the study, consisting of 10 non-CRS controls, 10 CRSsNP and 26 CRSwNP patients. Demographic and clinical characteristics of the patient groups are reported in Table 1. CRSwNP patients were more likely to be males and had a higher tissue eosinophil score compared to controls. There were significant differences in median levels for total serum IgG, IgG1, IgG2, IgG4 and IgE, but not IgG3 between controls, CRSsNP and CRSwNP patients (Table 1 and Figure 1). Three factors (explaining a total variance of 83.3%) were retained following principal component factor analysis of serum immunoglobulin levels. Interpretation of these factors based on factor loadings were: Factor one captured total IgG and the most abundant subgroups IgG1 and IgG2; factor two captured IgG4 and IgE; and factor three captured IgG3 (Appendix A). A scatterplot of observed serum immunoglobulin levels, by group, is depicted in Figure 1. A scatterplot of individual patient scores for the first two factors demonstrates that lower scores for both factors predominantly occurred in controls, and higher factor two scores predominantly occurred in CRSwNP patients, which is consistent with the distributions of individual immunoglobulin levels reported in Figure 1. Differences in factor scores between patient groups were analyzed by linear regression with bootstrapped standard errors (Appendix A). An additional analysis was also performed with adjustment for covariates. This included age, gender, asthma and oral steroids within the last 3 months. Factor 1 scores (Total IgG/IgG1/IgG2) were higher in CRS patients vs. controls (*p* < 0.001), but were not different between CRSwNP and CRSsNP patients (*p* = 0.57). These conclusions were unaffected by covariate adjustment. In contrast, factor two scores (IgG4/IgE) were increased in CRSwNP patients (*p* = 0.006). However, this relationship was attenuated by the inclusion of covariates (*p* = 0.54), which is indicative of confounding between variables. Tissue eosinophilia was more pronounced in patients with asthma (*p* = 0.030, exact median test), and both tissue eosinophilia and asthma were increased in CRSwNP patients as expected (Table 1). Interestingly, serum IgG4 levels were more strongly associated with asthma (*p* = 0.038, exact median test) and tissue eosinophilia (Spearman’s rank rho = 0.51, *p* = 0.016) than either IgE levels or factor three (data not shown). The presence of antibodies to 12 autoantigens was also tested in the serum of these patients. All three positive controls were positive for at least one autoantibody. No autoantibodies were detected in the sera of any of the CRS patients. We then determined total IgG levels in nasal polyp tissue or non-polyp mucosal tissue from CRSwNP and mucosal tissue from CRSsNP and non-CRS control patients. Nasal polyp IgG levels were significantly increased compared to CRSsNP (*p* = 0.0005) and non-CRS control mucosa (*p* < 0.0001). Within CRSwNP, nasal polyp IgG levels appeared to be higher than that of non-polyp mucosa, but this did not reach statistical significance (*p* = 0.059). IgG levels in CRSwNP mucosal tissue were significantly higher than in non-CRS control (*p* = 0.021) (Figure 1). Next, we determined whether correlations existed between serum and tissue IgE and IgG4 levels by using immunohistochemical analysis on matched tissue sections from the three patient groups (controls, n = 9; CRSsNP, n = 6; CRSwNP, n = 18). The intensity of IgE and IgG4 was measured at different areas of submucosa and compared amongst patient groups. IgE intensity in CRSwNP tissue was significantly increased, compared to controls (*p* ≤ 0.015) and CRSsNP (*p* ≤ 0.017) patients (Figure 1). In addition, IgG4 expression was significantly increased in CRSwNP in comparison to controls (*p* ≤ 0.0001) and CRSsNP (*p* ≤ 0.002) patients (Figure 1). There were no significant correlations between tissue and serum IgE or IgG4 levels. This study confirmed published observations of increased tissue IgG, IgE and IgG4 levels in CRSwNP patients compared to controls [3,4]. In contrast with those studies however, and potentially due to differences in the sensitivity of analytical assays used, our results also indicate increased systemic levels of total IgG/IgG1/IgG2 in CRS patients vs. controls and increased IgG4/IgE levels in CRSwNP vs. controls. This latter increase might at least in part be due to confounding factors with serum IgG4 levels strongly associated with asthma and tissue eosinophilia. A polyclonal increase in immunoglobulin levels, as seen in our study, is usually related to immune activation associated with autoimmune diseases or infection. This study failed to demonstrate the presence of serum anti-nuclear antigens (ANA’s) in any of the patients tested, making a diagnosis of autoimmune ANA mediated sinus inflammation unlikely. Chronic relapsing infections and mucosal biofilms involving different pathogens such as *S. aureus* and *P. aeruginosa* are, however, key elements in the pathophysiology of CRS and might underlie the increased immunoglobulin levels seen in this study. Increased levels of serum and tissue IgG4 are typically found in IgG4-related disease (IgG4-RD), characterized by lesions showing specific immunopathologic features, such as lymphoplasmacytic infiltration, storiform fibrosis, and obliterative phlebitis [6]. Whilst the pathophysiology of IgG4-RD is unknown, IgG4-RD patients also frequently suffer from allergic diseases, including asthma and atopic dermatitis and the disease can affect multiple organs, including the lungs and paranasal sinuses. Serum immunoglobulin IgG4 elevation has been associated with several pathological conditions other than IgG4-RD however, including in the context of relapsing infections, autoimmune diseases, cancer and cystic fibrosis (CF) [7]. In a cohort of 165 CF patients, 26% had increased serum IgG4 levels in association with elevated levels of IgG1, IgG2, IgE but not IgG3, (a uniquely potent immunoglobulin that triggers effector functions against a range of pathogens) precisely mirroring our results. Interestingly, those patients also had an increased prevalence of *Staphylococcus aureus* colonization [8]. Furthermore, whilst serum immunoglobulin levels were normal in their patient cohort, Van Zele et al showed that CRSwNP tissue with specific IgE to *S. aureus* enterotoxins (SAEs) had significantly higher concentrations of IgG and IgE and a significantly higher fraction of IgG4 [3,4]. Together with our study results, these findings might support the hypothesis of a potential role for *S. aureus* in the polyclonal increase in serum immunoglobulins with elevated IgG4/IgE levels in CRSwNP patients marking tissue eosinophilia and asthma. Further research is needed to measure IgG4/IgE levels in a larger CRS patient cohort and in relation to atopy and S. *aureus* colonization.

## Figures and Tables

**Figure 1 pathogens-09-00828-f001:**
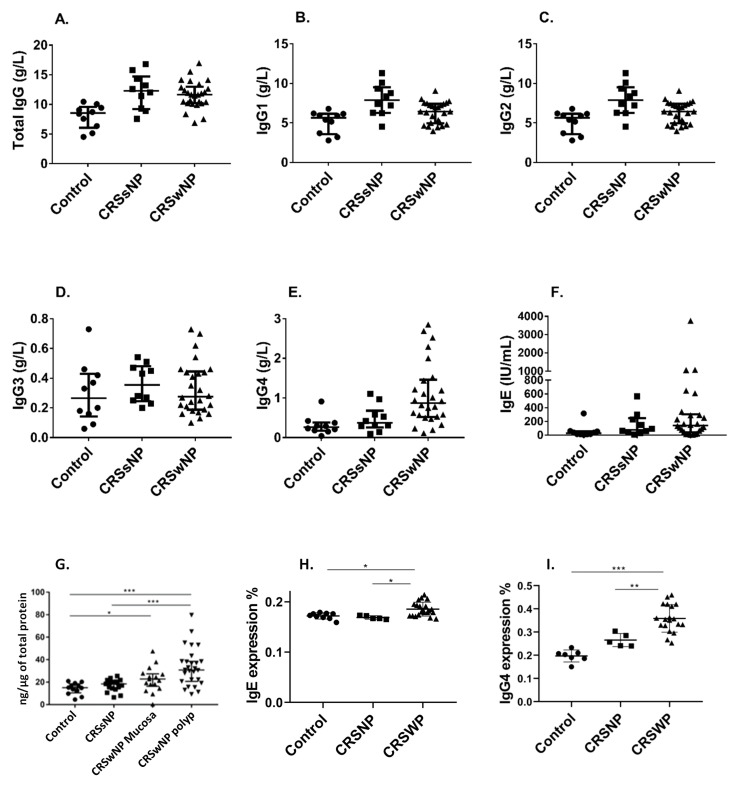
Serum immunoglobulin Total IgG (**A**) IgG (**B**), IgG2 (**C**), IgG3 (**D**), IgG4 (**E**), IgE (**F**) levels and tissue total IgG levels by patient groups and tissue type (**G**) in controls (n = 10), chronic rhinosinusitis patients without nasal polyps (CRSsNP, n = 10) and chronic rhinosinusitis patients with nasal polyps (CRSwNP, n = 26). IgE (**H**) and IgG4 (**I**) expression in CRSwNP, CRSsNP and non-CRS control patients. The average optical density (AOD) of positive IgE and IgG4 immunohistochemical staining was measured using Image J v1.52a (NIH, USA). * *p* < 0.05, ** *p* < 0.01, *** *p* < 0.0001.

**Table 1 pathogens-09-00828-t001:** Demographic, clinical features and serum immunoglobulin levels of patients included in the study.

Parameter	Control	CRSsNP	CRSwNP	*p * ^1^
N	10	10	26	
Age: mean (sd)	54 (9)	45 (19)	53 (14)	0.30
Females	6/10 (60%)	5/10 (50%)	5/26 (19%)	0.041
Asthma	2/10 (20%)	4/10 (40%)	16/26 (62%)	0.083
Oral steroids within last 3 months	3/9 (33%)	5/10 (50%)	12/24 (50%)	0.71
Local steroids within last 3 months	2/9 (22%)	7/10 (70%)	13/24 (54%)	0.11
>1 Previous operations	0	1 (19%)	2 (20%)	
Tissue Eosinophilia: median (IQR) ^2^	0.7 (0, 1.6)	0.5 (0, 1.3)	42.4 (15.2, 54.2)	0.002
Serum Immunoglobulins: median (IQR)				
*IgG (g/L)*	8.56 (3.06)	12.35 (5.02)	11.65 (2.63)	0.001
*IgG1 (g/L)*	5.65 (2.45)	7.88 (3.05)	6.44 (2.38)	0.018
*IgG2 (g/L)*	2.07 (0.74)	3.36 (1.92)	3.96 (2.22)	0.010
*IgG3 (g/L)*	0.27 (0.26)	0.36 (0.22)	0.28 (0.25)	0.92
*IgG4 (g/L)*	0.26 (0.19)	0.37 (0.29)	0.87 (0.91)	0.002
*IgE (IU/mL)*	26 (40)	78.5 (194)	141.5 (246)	0.010

^1^ Comparisons across the three groups were performed by ANOVA, with bootstrapped standard errors, Fisher’s exact test, or an exact median test for immunoglobulin levels. ^2^ Count per High Power Field (0.035 mm^3^). Eosinophil counts were obtained for s subgroup of patient (5 controls, 5 CRSsNP and 12 CRSwNP). CRS: Chronic Rhinosinusitis; CRSsNP: CRS without nasal polyp; CRSwNP: CRS with nasal polyp.

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
