# Peer review of "Increased Serum IgG4 Associates with Asthma and Tissue Eosinophilia in Chronic Rhinosinusitis Patients"

_pathogens, 2020, doi:10.3390/pathogens9100828_

Round 1
Reviewer 1 Report
In a descriptive study the authors examined the impact of thepreence of SRC and non-RC on the levels of total gG,subclasses of IgG, IgE and IgA in nasal tissue and erum samples. They also evaluated immunoglobulin levels in relation to disease severity, phenotype, histopatholgy and comorbidities.
Major Points
- The authors measured IgE in both blood and nasal tissue. Local and sytemic IgE levels are, at least in part, asssociated with the presence of atopy. However, atopy was not asessed either by skin prick test or by specific IgE for more frequent allergens present in the geographical area. Moreover, IgE agaisnt S. aureus was not measured despite it is well known that high levels of s. aureus polyclonal IgE are frequently detected in CRS patients. The two limitations of the study hould be mentioned in the dicussion part of the manuscript.
- The size ofthe samples was small and consequently the statistical power of the study is low to detect significant differences. This limitaion should also be mentioned.
Minor Points
- Characterization of CRS a eosinophilic (Type 2) and neutrophilic (Type 1) is an oversimplification of what is usually found in the histology of CRS samples. A recent study questioned this simplistic approach to characterize CRS.
Delemarre T, el al. J Allergy Clin Immunol 2020;september 16:S0091-6749(20)31275-6 (on line)

Author Response
Main points
- The authors measured IgE in both nasal tissue and blood. Local and systemic IgE is, at least in part, associated with the presence of atopy. However, atopy was not assessed by prick test or by specific IgE for more frequent allergens. IgE against S. aureus was not measured despite it is well known that is frequently detected in CRS. Both limitations of the study should be mentioned in the discussion part of the manuscript.
The limitations have been addressed (last sentence of the discussion). In fact, we did evaluate the relation of atopy (skin prick tests) with immunoglobulin levels but the atopy status was not available for sufficient patients.
- The size of the samples was small and consequently the statistical power is low to detect some differences. This limitation should be mentioned.
This has been addressed (last sentence of the discussion).
Minor points.
- The characterization of CRS as eosinophilic (type 2) and neutrophilic (type 1) has been questioned. A recent study shows that histological classification is more complex than simply dividing CRS into two well-defined types.
This has been modified.
Reviewer 2 Report
This study was to investigate “Increased Serum IgG4 Associates with Asthma and Tissue Eosinophilia in Chronic Rhinosinusitis Patients”. The authors hypothesized potential role for S. aureus in the polyclonal increase in serum immunoglobulins with elevated IgG4/IgE levels in CRSwNP patients marking tissue eosinophilia and asthma.
There are several concerns in this manuscript.
- When the title of the manucript is “Increased Serum IgG4 Associates with Asthma and Tissue Eosinophilia in Chronic Rhinosinusitis Patients “. the authors should divided their subjects into the controlgroup, CRS patients without asthma and tissue eosinophila and CPR patients with asthma and tissue eosinophila.
- The level IgG3 did not increase in CRS patients as IgG1 aand IgG2, the authors should give a more detailed discussion of the result.
- Similarly, the authors hypothesized the potential role for S. aureus in the polyclonal increase in serum immunoglobulins with elevated IgG4/IgE levels in CRSwNP patients marking tissue eosinophilia and asthma. However, the current evidences are too weak to support the hypothesis.
- Finally, when the authors divided their subjects into the control group, CRS patients without polyps and CPR patients with polyps, they did not emphasize the role of polyps in the whole paper.
Author Response
- When the title of the manucript is “Increased Serum IgG4 Associates with Asthma and Tissue Eosinophilia in Chronic Rhinosinusitis Patients “. the authors should divided their subjects into the control group, CRS patients without asthma and tissue eosinophila and CPR patients with asthma and tissue eosinophila.
The title reflects the main finding of the paper. This was identified using advanced statistical analysis techniques (i.e. regression analysis with adjustment for covariates). Dividing patients into controls, CRS without asthma and without tissue eosinophilia and CRS patients with asthma and with tissue eosinophilia was also done and analysed, however, results were less strong due to a reduction of power with the exclusion of those patients that had tissue eosinophilia but no asthma. As the reviewers correctly indicate, the criteria for dividing the different CRS phenotypes (and endotypes) into CRSsNP and CRSwNP and CRS with and without eosinophilia and with and without asthma is not clear so a more sophisticated analysis is needed to determine the main factors that determine a specific phenotype. This has been carried out in this paper.
- The level IgG3 did not increase in CRS patients as IgG1 aand IgG2, the authors should give a more detailed discussion of the result.
We have discussed this (taking into account the limited number of words that we can spend given the letter format).
- Similarly, the authors hypothesized the potential role for S. aureus in the polyclonal increase in serum immunoglobulins with elevated IgG4/IgE levels in CRSwNP patients marking tissue eosinophilia and asthma. However, the current evidences are too weak to support the hypothesis.
We have modified this part.
- Finally, when the authors divided their subjects into the control group, CRS patients without polyps and CPR patients with polyps, they did not emphasize the role of polyps in the whole paper.
In the analysis of tissue immunoglobulin levels, CRSwNP patient’s tissues comprised nasal polyps and mucosal tissue. This has been described in the relevant section in the manuscript (Figure 1G).
Reviewer 3 Report
In Letter to the Editor Authors present results of their prospective study on immunoglobulin levels in serum of patients with chronic rhinosinusitis with and without nasal polyps. Serum IgG4 levels are found to be correlated to asthma and tissue eosinophilia without detection of systemic ANA. The increased relapsing infections with S.aureus and P.aeruginosa seem to be key elements in pathophysiology of chronic rhinosinusitis which might underline the increased Ig levels.
Author Response
In Letter to the Editor Authors present results of their prospective study on immunoglobulin levels in serum of patients with chronic rhinosinusitis with and without nasal polyps. Serum IgG4 levels are found to be correlated to asthma and tissue eosinophilia without detection of systemic ANA. The increased relapsing infections with S.aureus and P.aeruginosa seem to be key elements in pathophysiology of chronic rhinosinusitis which might underline the increased Ig levels.
Thank you, whilst further research is needed to relate infection with specific pathogens and Ig levels, this certainly seems to be the case.
Round 2
Reviewer 1 Report
The authors have addressed the points raised in the revised manuscript
Reviewer 2 Report
No further question